# Fresh-Water Mollusks as Biomonitors for Ecotoxicity of Nanomaterials

**DOI:** 10.3390/nano11040944

**Published:** 2021-04-08

**Authors:** Natalia Abramenko, Petr Mashkin, Sergey Volkov, Vladimir Olshanskiy, Leonid Kustov

**Affiliations:** 1N.D. Zelinsky Institute of Organic Chemistry, Leninsky Pr. 47, 119991 Moscow, Russia; natalimsu@gmail.com; 2A.N. Severtsov Institute of Problems of Ecology and Evolution, RAS, 119071 Moscow, Russia; pmashkin@yandex.ru (P.M.); mendur@mail.ru (S.V.); vmolsh@yandex.ru (V.O.); 3Chemistry Department, Moscow State University, Leninskie Gory 1, Bldg. 3, 119991 Moscow, Russia

**Keywords:** bioindicators, mollusks, heart rate, ecotoxicity, inorganic nanoparticles

## Abstract

The use of different nanoparticles (NPs) is growing every year since discoveries of their unique properties. The wide use of nanomaterials has raised concerns about their safety and possible accumulation in the aquatic environment. Mussels are being considered as one of the most suitable organisms for bioaccumulation monitoring. Within our study, we focused on developing the method that can be applied in field studies of ecotoxicity and can be nondestructive and informative at early times of exposure, while at the same time being based on changes of physiological parameters of fresh water mussels. The changes in the cardiovascular and neural systems of mollusks (*Anodonta anatina* and *Unio tumidus*) were measured as biomarkers of toxic effects. Different monometallic and bimetallic NPs, silicon NPs with various ligands were applied as test substances. Changes in cardiovascular and neural functions were in good correlation with accumulation tests for all tested NPs.

## 1. Introduction

Nowadays metal nanoparticles (NPs) are considered as promising materials for targeted drug delivery and cancer treatment [1]. Due to significantly high capacity and low discharge potential, the NPs are interesting for electronics and optoelectronic applications and in energy storage systems.

The rates of the development of such materials are much higher than the possibilities of their detailed characterization and monitoring in terms of the toxicity and fate in the environment. It was pointed out that studies of NPs related to their effects on environment should be carried out to understand possible risks of their wide use [2]. Many NPs demonstrated genotoxicity and evidence of DNA damage towards different cell types [3,4,5]. For instance, Si-based particles have also been explored as a gene carrier, demonstrating promising results to effectively transfect and deliver genetic material. The effects of silver and copper NPs have been described in numerous reviews on the biological effects of NPs. In particular, both copper and silver NPs have been shown to cause gill pathology in Danio rerio, their respective LD_50_/48hrs was 4.2 and 2.9 mg/L [6]. Sourav et al. reported an important role of the surface charge of Si-core NPs in determining their cytotoxicity towards macrophage NR8383 cells [7]. According to the results obtained by Ahire et al. [8], carbohydrate capped Si NPs can be internalized by cancer cells as well as non-cancer cells. It was shown in several in vivo studies [9,10] that NPs are mainly accumulated in liver and spleen for Wistar Han rats and Nude mouse. At the same time, the authors determined the fast clearance of the NPs from the bodies of tested animals. Safety of NPs and their toxicological properties depend on the particles’ surface properties, their structure and size characteristics [11]. The nanotoxicity tests are however being usually performed on model cell lines (in vitro tests). It should be noted that nanotoxicity tests based on cellular lines and organelles do not provide sufficient information about the toxicological effect on the entire organism [12].

Nevertheless, the acute and long-term toxicity of NPs to organisms and environment are not completely understood. Aquatic ecosystems are the ultimate receiver of various contaminants. The world practice is based on the use of fishes, fresh- and salt-water mollusks, crustaceans, and sea weeds [13,14,15,16]. Mollusks play a very important role in water ecosystems as filtrating organisms. All the pollutants gradually accumulate in these organisms. Their level of health is a biological indicator of the load on the ecosystem. According to Rocha [17], the studies about the toxic effects of engineered nanomaterials in bivalves were carried out mainly with seawater species compared to freshwater ones.

Researchers usually study the accumulation of pollutants in organism tissues, the damage of biochemical processes in the organs of mollusks, the time of mortality of organisms at different pollutant expositions [7]. However, before the death of the organism and even before any serious damage to a mussel health, the physiological parameters of organisms start to change. The cardiovascular and neural systems are first to react to external hazards and to any of the allowable limits of pollution. Under any pollutions of the media, mollusks close the valves, their cardio rhythms are destabilized and slowed down until the complete stop of the cardio activity, which does not yet mean the death of the organism [13,18]. Therefore, taking into account that the cardiovascular and neural functions are most important functions of the hydrobionts, the main goal of this study was the development and approbation of the methods for assessment of the toxicity of different types of NPs using cardiovascular activity of fresh-water mollusks *Anodonta anatina (A. anatina)* and *Unio tumidus (U. tumidus)* as biosensors. The cardiovascular signal was observed for two types of mussels with and without exposure to NPs. Different Me and bimetallic NPs, silicon NPs with organic ligands were applied as tested materials. The uptake of NPs was investigated prior and after the end of the experiment.

This study aimed at exploring the usefulness of fresh-water mussels as perspective biomonitors for different pollutants. The method was applied to reveal the changes following the exposure to NPs in fresh-water mussels and provided insights into the action of silver, copper, bimetallic and silicon NPs towards mussels.

## 2. Materials and Methods

### 2.1. Tested Nanoparticles

Commercial silver and synthesized Si NPs were used for ecotoxicity tests. Colloidal solution of 9–15 nm silver NPs at the concentration of 280 mg/L of silver, stabilized with sodium bis(2-ethylhexyl) as a surfactant, at the concentration of 27 g/L were provided by JSC Nanoindustry Co. (Moscow, Russia).

Copper and bimetallic Cu-Ag NPs with core-shell structure as a colloidal solution was prepared by the radiation-chemical reduction procedure [19,20]. Synthesis of Si NPs was carried out according to the methods described elsewhere [21,22]. As a result, solutions of the dispersion of Si-core NPs with a different organic shell were obtained.

UV–vis absorption spectra were recorded using a Varian Cary 100 spectrophotometer with cell Peltier accessory (Varian, Inc., Palo Alto, CA, USA). A Carl Zeiss LEO912 AB OMEGA microscope (LEO Electron Microscopy Inc., Thornwood, NY, USA) was used to obtain transmission electron microscope (TEM) images of the tested NPs. For TEM analyses, samples were prepared by placing one or two drops of the nanoparticle solution onto the formvar resin coated copper grid and drying it in air at room temperature. The selected area electron diffraction (SAED) was also carried with the LEO912 AB OMEGA microscope (LEO Electron Microscopy Inc., Thornwood, NY, USA) with the voltage of 100 kV. Carl Zeiss NVision 40 scanning electron microscope (Carl Zeiss Group, Oberkochen, Germany) was used to obtain scanning electron microscope (SEM) images of the NPs. Nanoparticle sizes were determined in the colloidal solution by dynamic light scattering (DLS). The measurements were carried out with a DelsaTMNano C particle analyzer (Beckman Coulter, Brea, CA, USA) at a wavelength of 658 nm using the Delsa Nano Software package.

The prepared NPs were characterized in the modes of the light and dark fields at the accelerating voltage of 300 kV by HRTEM (TITAN 80-300, FEI Ltd., Hillsboro, OR, USA) with the correction of the spherical aberration. In the case of the dark-field experiments, a large-angle circular detector of scattered electrons (HAADF) was used in order to form the Z-contrast. The X-ray energy-dispersion microanalysis (EDAX, FEI Ltd., Hillsboro, OR, USA) was applied to determine the chemical composition of the samples. Processing and interpretation of the experimental data were performed using the Digital Micrograph (Gatan, Pleasanton, CA, USA) and TIA (FEI Ltd., Hillsboro, OR, USA) methods.

### 2.2. Toxicity Assessment Using the Bivalve Mussels

In our experiments on the toxic effect of various NPs colloidal solutions, we used the most common species of filtrating bivalves from the Oka River basin, the swollen river mussel (*A. anatina* and *U. tumidus*), which dominates the macrobenthos in terms of both abundance and biomass. For toxicity experiments, animals of the length 8–12 cm at the age of 9–12 years were selected. Thus, animals should exhibit about the same filtration ability.

The control and experimental groups of mollusks were kept in cylindrical containers. The volume of water was 20–30 L. Temperature was 20 °C. The photoperiod included 12 h of dark and 12 h of artificial light. In order to create identical conditions for every animal, mussels were placed into oval slots of a special platform. Each mussel was positioned vertically, with its posterior siphon bearing end directed upwards and the “foot” directed downwards. Water was aerated with air compressors. Additionally, water was mixed using a submersible pump to prevent possible sedimentation of NPs. The animals received no food one week before the experiments and during the experiment.

In most of the researches, authors do not pay attention to the level of initial pollution and the level of accumulation of various pollutants in the tissues of mollusks before experiments. Certainly, mussels might be already adapted to a certain background of pollution. Thus, before the tests with NPs analyzes of the content in tissues of mussels of the main traditional pollutants for a water basin were carried out.

### 2.3. Registration of Cardio Rhythms and Processing of Signals

In order to precisely determine the instant of death and to estimate the toxic effects of NPs to bivalves, appropriate physiological parameters (heart rate, opening/closing of the shell) were used. To record the rhythmic activity, we used the original 6-channel optocardiograph designed for registration of low-frequency physiological rhythms [23].

The optocardiograph allows one to simultaneously register, visualize, and save on a computer the heart rates of up to six mollusks. IR sensors were attached on the right shell on a ventricle of the heart zone without any injury (Figure 1).

The optimal site for gluing the sensor to the shell was selected individually for each studied mussel. The criteria for selection were the maximum amplitude of the obtained signal and the maximum similarity of the signal shape to the typically observed shape for the species. CNY70 infra-red sensors were tightly secured on the shells using a waterproof glue. During this procedure, which lasted 5–10 min, the mussel was held out of water, which had no adverse effect.

The sensor registers the changes in the optical density as a result of the flow of blood in the area of the ventricle heart.

Before the experiment, the shape of the cardiogram was registered and averaged for each individual mollusk during one day in order to derive the individual control cardiogram. The operator forms a typical image form of the signal. The computer program scans the record, finds forms, which have correlation on the 75% level comparable with the typical image form and calculates the time intervals between the beats. Samples of 6 cardio signals are presented in Figure 2.

This data processing brings us information about the changes in time of physiological parameters of cardio activity under NPs influence on the organisms. Under normal conditions, mussels close shells and show a much slower pulse one-two times in a day for 3–4 h. For the majority of *A. anatina* and *U. tumidus* mussels under normal conditions, the time between heart beats varies from 4–7 to 8–10 s, respectively.

### 2.4. Ecotoxicity Studies

Each experimental group consisted of 10 mussels; in each group, two mussels carried IR sensors to monitor their heart rate. The NPs samples, stabilizer and organic solvent were tested for the toxicity towards the freshwater mussels.

The level of closure of shells was controlled visually. The instant of the death is determined on a cardiogram (the start of a chaotic distribution of the intervals between the pulses and, as a result the impossibility of the regeneration of the normal shape of the cardiogram). After that, the reaction of adductors during the external attempts to open the valves should be tested. If the cardio rhythms are absent and the valves can be opened easily, the animal is considered not alive.

The study was performed with Ag NPs and silver ions (AgNO_3_) of different concentration (0.1; 0.75 mg/L) towards *U. tumidus.* We evaluated the toxicity of the Ag NPs dispersion, which contains NPs of silver stabilized by sodium sulfosuccinate.

Experiments with Cu and Cu@Ag NPs were conducted using three groups of animals: Group 1, control; Group 2 was placed into the solution containing copper NPs; Group 3 was placed into the solution with NPs containing approximately equal parts of copper and silver (Cu@Ag). In both solutions, the concentration of NP was 10^−8^ mol/L.

The ecotoxicity of the following samples of NPs was studied previously [24]:

Si_n_-(C_4_F_9_)_m_ (the nanoparticles contain C_4_F_9_ groups at their surface);

Si_n_-(CH_2_)_m_ (the nanoparticles contain imidazol carbene groups at their surface denoted as =CH_2_);

Si_n_-(C_4_H_9_)_m_ (the nanoparticles contain C_4_H_9_ groups at their surface).

The Si NPs were initially stabilized in monoglym (1,2-dimethoxyethane, CH_3_OCH_2_CH_2_OCH_3_)

The experiments were carried out at the same concentration of silicon NPs of 15 mg/L by diluting the initial concentrated solution in monoglym with tap water.

### 2.5. Preparation of the Samples of Gills and Liver and Elemental Analysis

The samples of the tissues (gills and liver) for the analysis for alien elements were taken right after the termination of the ecotoxicity experiment with each group of mollusks. The samples of gills were taken from both the right and left internal gills. The liver was used as a whole. The samples of tissues were dried in a microwave oven at 103 °C for 3 h. The decomposition of tissues was carried out in acids using a standard procedure. Accumulation of Si in tissues was measured with a Perkin Elmer atomic absorption spectrometer (AA240-FS and AA240Zgta120 instruments, Waltem, MS, USA). Wet combustion was used to prepare the samples for analysis.

## 3. Results and Discussion

### 3.1. Nanoparticles Characterization

Prior to the toxicity tests, the tested samples were characterized. All tested NPs were spherical with a narrow size distribution. As examples of TEM results, microphotograph and microdiffraction of Ag NPs are shown in Figure 3.

TEM data demonstrate that all particles have a narrow size distribution and a spherical shape (Appendix A). Typical electron microdiffraction patterns indicate the crystalline structure for all tested NPs (example demonstrated for Ag NPs).

Size distributions and optical spectra of Cu and Cu@Ag NPs are shown in Figure 4.

The solutions of Cu and Cu@Ag NPs manifest typical UV-bands with maxima at 570 nm and 350 nm. The data on characteristics of tested NPs are summarized in Table 1.

These data clearly show that the samples are characterized by a very narrow size distribution, although the average size of the Cu@Ag NPs is about twice larger as compared to pure Cu NPs. The mean median particle size of Si NPs is between 2 and 4 nm depending on the preparation. Data on the size distribution of tested NPs estimated by TEM are in a good agreement with results of DLS measurements.

### 3.2. Bioaccumulation of HM in the Tissues of Mussels

Because the mussels might have already some elements accumulated in their tissues, we assessed the initial content of some metals in the liver and gills of the control group of 10 mussels prior the start of experiments (Table 2). Since we focused on pollutant metals, we chose the site characterized by moderate levels of pollution of water and sediment by heavy metals (HM).

At the same time, we found a certain amount of silicon, silver, and copper in the liver for the control group of mollusks. Similar results were obtained for gill accumulation (data are not shown). As we used mollusks taken from natural water objects, it should be taken into account that animals already accumulated some quantity of different chemicals and have adapted to a certain level of pollution of the ambient medium. This factor, by a non-predictable way, may affect the rate and level of the development of physiological reactions on any experimental impacts.

### 3.3. Approbation of the Method Using Different NPs

Different NPs were taken for approbation of the methods. The NPs samples, stabilizers (docusate sodium salt, SDS) and the solvent, 1,2-dimethoxyethane (monoglym), were also tested towards mussels. We evaluated the toxicity of the Cu, bimetallic Cu@Ag NPs and commercial Ag NPs, which contain silver NPs stabilized by sodium sulfosuccinate, towards fresh-water mussels—*U. tumidus*.

### 3.4. Approbation with Silver NPs and Silver Ions

During the first day of the experiment with Ag NPs and *U. tumidus*, all the test groups displayed a slowdown of the heart rate (intervals between heart pulses, 8–10 s), but the valves were partially open and filtration continued with a slower rate. The amplitude and shape of the cardiograms changed. After the first day, mussels of all groups closed their shells, and their heart rates appeared to have “died”. After the slight stimulation by pressing or knocking at the shell, all the groups restored their heart signal amplitudes. At the end of experiments, the heart rates recovered at all concentrations with exception of the high concentration (0.75 mg/L), with the latter only half of the animals could restore their heart signal amplitudes. In our experiments with the Ag NPs, mortality was recorded among experimental animals exposed to that concentration of silver for seven days. Moreover, the data show that the stabilizer by itself is toxic, almost as highly as the Ag NPs themselves (Figure 5 and Table 3).

At high concentrations, the toxic effect of the stabilizer is similar to the toxicity of Ag NPs dispersion. For lower concentrations, silver dispersions appeared more toxic than the stabilizing agent. In dilution solution higher toxicity can be related with silver ions release from surface of Ag NPs. Silver ions demonstrated higher toxicity towards *U. tumidus* than Ag NPs (Table 3).

We evaluated the accumulation of Ag NPs in contrast with silver ions accumulation in mussel tissues. We conducted a series of experiments with AgNO_3_ to study the dynamics of silver accumulation in the organs of *U. tumidus* mussels in the absence of a stabilizer and Ag NPs.

The concentrations of ionic silver accumulated in the tissues turned out to be significantly higher than in our experiments with the same concentrations of Ag NPs (Figure 6). Given the same time of exposure to AgNO_3_ solutions, the mean concentration of silver in the liver and gills increases monotonously with its increasing concentration in the medium. Accumulation of Ag (total) in gills was higher than in liver both for ions and for NPs.

### 3.5. Approbation with Silicon NPs with Different Organic Ligands

Other results were obtained with Si NPs and *A. anatina*. Since Si NPs were prepared in the solution of monoglym, it was worth to check for the toxicity of the mere organic solvent. The exposure concentrations were 15 mg/L for all tested samples, and they were used to estimate the LD_50_ values of Si NPs solutions and solvent.

Experiments were carried out for 7–8 days for all tested NPs, for the most toxic Si_n_-ligands tests were stopped due to death of animals before the end of the experiment. Figure 6 shows histograms of times between the beats and cardiac cycle statistics on third day of the experiment for Si NPs (48-h incubation). The distribution for the different forms of Si NPs is shown with different color. The 50% mortality for the most toxic Si NPs was observed on the third day of experiments. The comparison of the ecotoxicity of different modifications of silicon NPs (Figure 7) revealed the following order of toxicity: C_4_F_9_ > CH_2_: > C_4_H_9._

Monoglym as a solvent also shows some toxicity or stress as instability in the heart rates during the experiments. The animals exposed to monoglym demonstrated the instability of cardio rhythms for two days, while the rhythms were stabilized at the third day. After six days, the periods of closing shells and changes in the heart rhythm revealed more often. After eight days, the heart rate normalized. Nevertheless, three animals were dead after eight days of exposure. This is an indication of a rather high toxicity of the clear monoglym solution.

The concentration-dependent accumulation was observed with silicon NPs [23]. The toxic effects of the studied Si NPs seem to originate from the nature of the organic groups forming the outer shell of the NPs. Of special concern is Si NPs with C_4_F_9_ groups containing fluorine. In case of Si NPs, the toxicity seems to be enhanced by the presence of fluorine in organic ligands. Nanoparticles Si_n_-(CH_2_)_m_, though being also toxic, behave quite differently compared to Si_n_-(C_4_F_9_)_m_. For 3 days, the heart rhythms are quite stable, and then the shape and amplitude of the cardio cycles change abruptly. The organism is trying to activate the defense mechanisms by closing the valves and by reducing the heart frequency. However, after 118 h, the death of 50% of animals is observed and the concentration of silicon in the liver reaches the same value as in the case of Si_n_-(C_4_F_9_)_m_ NPs. This accumulated concentration seems to be lethal for *A. anatina*.

The sample of silicon NPs with the outer shell composed of C_4_F_9_ groups turned out to be the most toxic among the tested silicon NPs with the highest rate of bioaccumulation. The lowest rate of bioaccumulation was observed for Si NPs with C_4_H_9_ stabilizing groups. The analysis data show that accumulation of silicon after the exposure to Si NPs occurs most effectively in the liver and gills in the case of the two types of the Si-core NPs modified with perfluorobutyl and carbene groups. It should be noted that the rate of accumulation in the case of C_4_F_9_ groups was higher, since the time of exposition was different (28 and 118 h, respectively). Accumulation of silicon in the gills and liver of mollusks in solutions containing Si-C_4_H_9_ NPs is not significant; the average values are close to the control group.

### 3.6. Approbation with Cu and Cu@Ag NPs

Mussels *U. tumidus* were more sensitive to the presence of bimetallic NPs than copper NPs. In the tested group with Cu@Ag NPs, the heart rhythm increases in eight hours after the start of the experiment (inter-pulse intervals, 4–5 s). After 22 h the average interval time returned to 6–8 s. In 46 h, we observed a sharp slowdown of rhythms (intervals, 12–14 s), the rhythms remained at the slow levels for the next 80 h. Towards the end of the experiment (168 h), the rhythmic activity restored its starting parameters (Figure 8). No death of animals was observed.

The concentrations of Cu and Ag in the mussel gills and livers are summarized in Table 4. Accumulation of silver is low (0.1–0.2 mg/kg), but it was statistically significant in comparison to the control group.

The data on the accumulation of copper and silver in tissues indicate that the increased concentration of metal NPs in the solution results in the increased concentration of silver in the liver and gills, even after shorter exposure times. The gills act as the primary target due to their large surface area, exposed to both dissolved pollutants and organic as well as inorganic particles of diverse origin. Nanoparticles can be small enough to enter interlamellar channels and inter-filament junctions and, possibly, pass cell membranes. Presumably, such particles can enter liver from the alimentary tract as well as via more complex routes. The dynamics of accumulation of NPs in the liver was different from those in the gills. This process is strongly affected by the duration of the contact with the polluted media. The data show that the concentration of copper and silver in the liver was greater than upon exposure to higher environmental concentrations, but the mollusks remain alive.

## 4. Conclusions

A non-destructive method to evaluate toxicity of pollutants towards fresh water mussels at early time of exposure was designed by using Me and Si NPs as test substances. Depending on the concentration, the effect of NPs on live organisms can be reversible or irreversible. It is first manifested by changes of physiological parameters, followed by structural changes. Structural changes require longer time to manifest than physiological ones. The organism tries to activate the defense by closing the valves and decreasing the heart rate. The demonstrated instability of cardio rhythms was in good correlation with the total accumulation of NPs in mussel tissues.

Based on the designed method, a correlation between the toxic effect of Si NPs on the heart rate of mussels and NP’s surface modification was shown. In the case of the most toxic modification of NPs, bradycardia was developed quickly, the neural-muscular transfer was blocked, and the mollusks could not close the valves. The sample of Si NP with fluorine was the most toxic due to the high rate of bioaccumulation in gills and liver of mussels. The silicon NPs functionalized with butyl groups were shown to be much less toxic. The data obtained show that the toxicity of the studied samples of silicon NPs, most likely, is related to the effect of the ligands.

Silver NPs appeared to be more toxic towards mussels than all tested Si NPs sample. Unlike Ag NPs, the target for Si NP is the lever but not the gills. The Ag NPs demonstrate lower toxicity than silver ions. Toxicity of Ag NPs could be related with their capacity to release free silver ions.

There is no clear correlation between the size and toxicity of tested NPs. It was shown that toxicity of NPs is correlated with nature of NPs, surface structure and toxic additives or solvent. The toxic behavior of NPs can be attributed to (1) the nature of NPs themselves, (2) stabilizing organic ligands, (3) the solvent, or (4) a combination of the two or three of these components. In a view of size characteristics, we have not found size dependent effects for tested NPs same as for shape of tested NPs.

Nanoparticles demonstrate a high rate of bioaccumulation in the tissues of mollusks and quickly affect the nervous and cardio-vascular systems of mollusks. As a general conclusion, additives and toxic solvents appear to play a significant role in NPs toxicity toward mollusks. The measurement of the heart rate and the valve opening provides an efficient method to evaluate the nanotoxicity by using mollusks. Thus, bivalve filtering mollusks are a convenient and perspective biological model for studying the effects of different pollutants on the physiological parameters of whole organisms.

## Figures and Tables

**Figure 1 nanomaterials-11-00944-f001:**
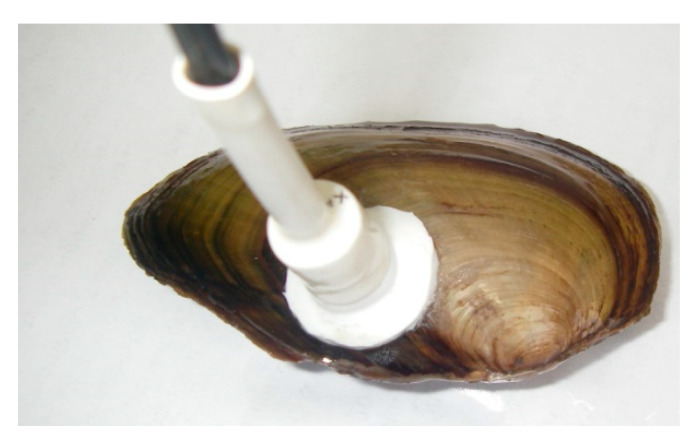
Sensor on the shell of *A. anatina* mussels.

**Figure 2 nanomaterials-11-00944-f002:**
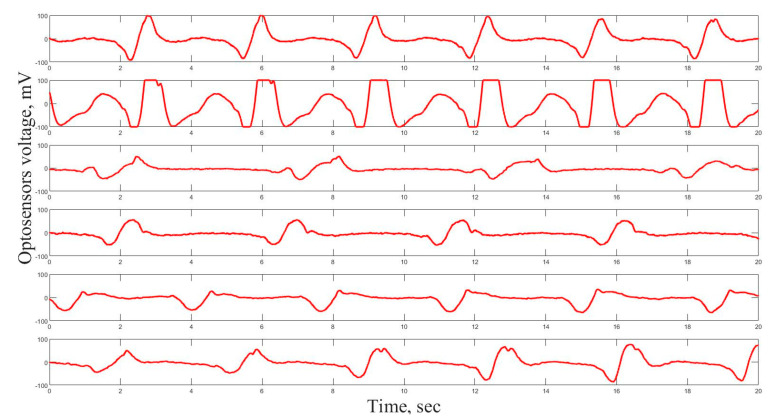
Samples of the cardiac signal of 6 mussels. The X axis shows the time of the experiment. The Y axis shows the amplitude of the heart beats in voltage of optosensors.

**Figure 3 nanomaterials-11-00944-f003:**
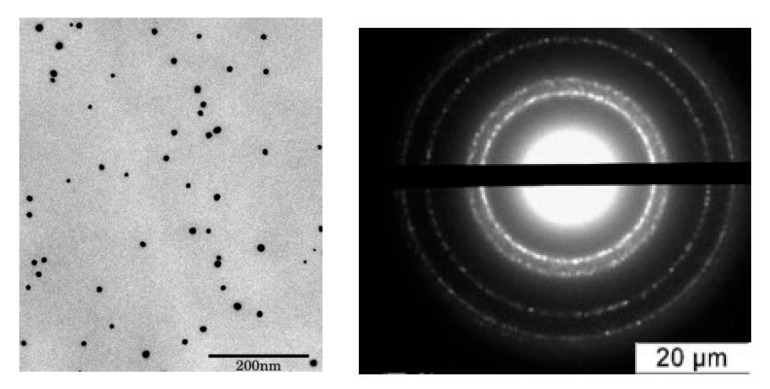
Transmission electron microscope (TEM) image (**left**) and microdiffraction of Ag nanoparticles (NPs) (**right**).

**Figure 4 nanomaterials-11-00944-f004:**
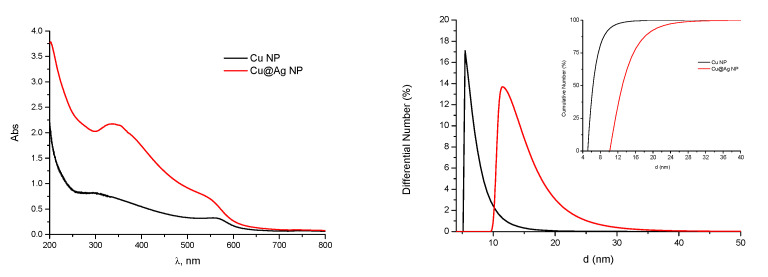
Optical spectra and size distributions of Cu and Cu@Ag NPs.

**Figure 5 nanomaterials-11-00944-f005:**
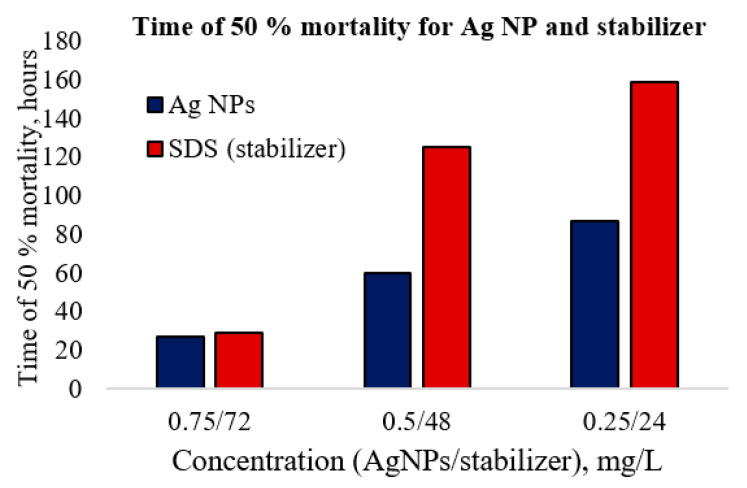
LC_50_ caused by silver dispersion and solution of the stabilizer at the same concentration as in the dispersion.

**Figure 6 nanomaterials-11-00944-f006:**
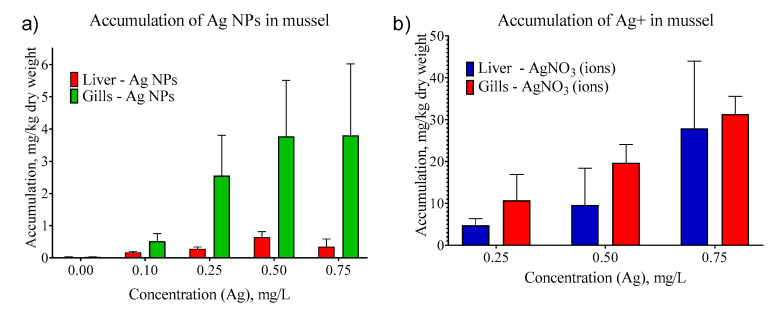
Accumulation of silver NPs (**a**) and silver ions (**b**) in *U. tumidus* tissues.

**Figure 7 nanomaterials-11-00944-f007:**
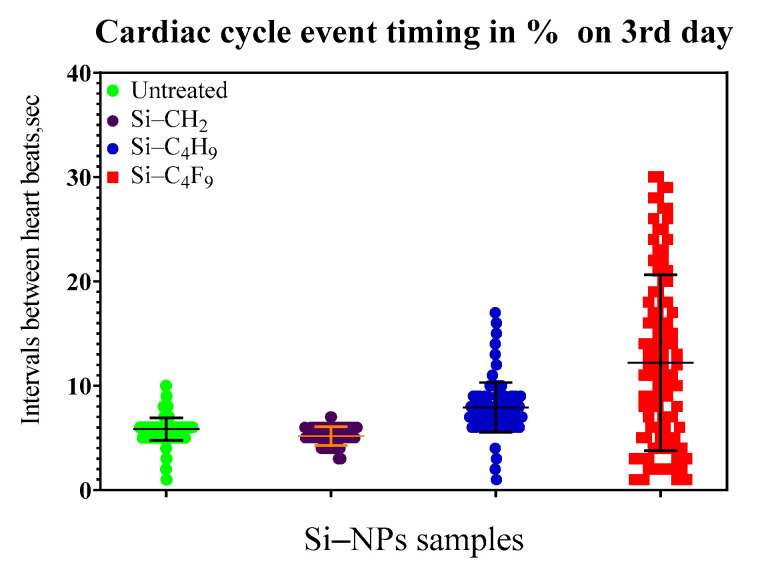
Distribution of time intervals between heart beats and cardiac cycle statistics (Y axis shows the time between the heart beats in the interval 0–35 s on the 3rd day of the experiment. X represent the sample).

**Figure 8 nanomaterials-11-00944-f008:**
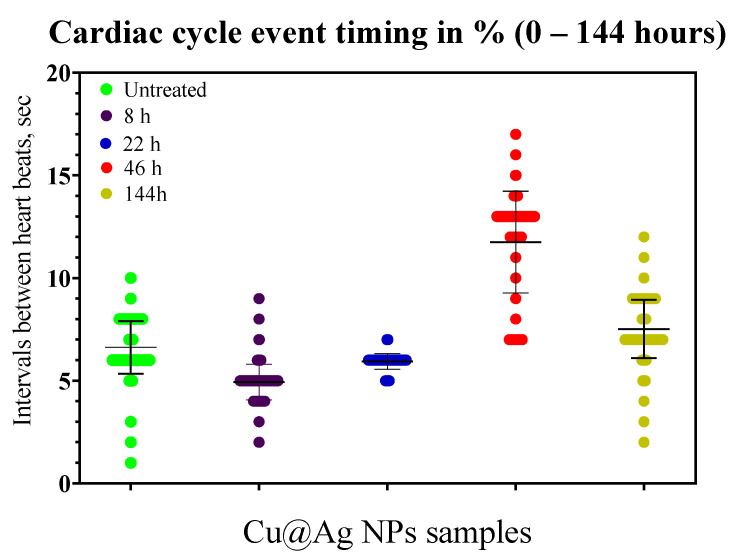
Distribution of time intervals between heart beats and cardiac cycle statistics for Cu@Ag NPs during the experiment (Y axis shows the time between the heart beats in the interval 0–18. X represents the time from the start of the experiment).

**Table 1 nanomaterials-11-00944-t001:** TEM data and size distribution for the tested NPs samples.

Sample	D (TEM), nm	SD	D (DLS), nm	SD
Si_n_-(C_4_F_9_)_m_	4	(1; 8)	4.5	(1.4; 8.9)
Si_n_-(C_4_H_9_)_m_	2.4	(1.5; 3.0)	2.2	(1.3; 3.7)
Si_n_-(CH_2_)_m_	1.8	(1; 3)	2.0	(1.1; 3.5)
Ag	11.2	(3.3; 18.7)	14.7	(10.5; 3.0)
Cu	6.0	(4.0; 8.0)	7.7	(5.1; 9.9)
Cu@Ag	13.9	(10.8; 17.0)	14.6	(10.6; 18.6)

**Table 2 nanomaterials-11-00944-t002:** Accumulation of heavy metals in mussel tissues (liver) in the control group (mg/kg dry weight).

Entry	Cu	Zn	Pb	Cd	As	Hg	Cr	Ni	Fe	Mn	Ag
1	22.5	258	1.50	1.76	6.71	0.040	1.80	3.32	0.61	1.23	0.038
2	24.1	289	1.12	1.34	11.20	0.078	1.20	2.67	0.87	2.13	0.023
3	13.7	310	0.75	1.00	9.43	0.083	2.10	3.21	1.34	1.87	0.012
4	20.3	232	0.81	1.32	14.23	0.046	2.22	3.94	2.02	2.23	0.009
5	19.4	190	0.92	1.20	11.90	0.078	1.89	2.90	1.92	2.13	0.021
6	15.3	213	1.62	0.83	10.35	0.080	1.43	2.34	1.56	0.89	0.027
7	16.1	304	1.31	2.50	9.30	0.130	1.62	4.10	0.93	1.45	0.028
8	22.1	216	2.00	3.00	7.80	0.080	1.10	3.12	0.76	1.91	0.023
9	21.1	190	1.00	0.50	8.90	0.060	1.37	3.47	0.56	3.20	0.030
10	17.3	520	0.89	1.00	2.40	0.087	2.60	3.03	2.35	3.35	0.032
Mean	19.2	272	1.19	1.45	9.22	0.076	1.73	3.21	1.29	2.04	0.024
SD	3.45	97.8	0.41	0.77	3.21	0.025	0.48	0.54	0.64	0.78	0.009

**Table 3 nanomaterials-11-00944-t003:** LD_50_ data and time of 50% mortality for the nanosilver and silver ion samples towards *U. tumidus*.

Sample	LD_50_, mg/L (Time)	Notes
Ag^+^ (AgNO_3_)	0.75 mg/L (14.4 h)	At first days, 50% of tested mussels died. The shells were partially opened.
Ag NPs	27 h (0.75 mg/L)	During the first two days, the shells were closed, then partially opened
	60 h (0.5 mg/L)	
	80 h (0.25 mg/L)	
	No death within 7 days (0.1 mg/L)	Observed during the total of 7 days; the rhythms were normal
Control	No death within 8 days	Observed during the total of 8 days; the rhythms were normal.

**Table 4 nanomaterials-11-00944-t004:** Accumulation of metals in mussel tissues in the control and treated group (mg/kg dry weight).

	Liver			Control (Liver)		Gills			Control (Gills)	
Entry	Ag	Cu	Cu Only	Ag	Cu	Ag	Cu	Cu Only	Ag	Cu
1	0.85	14.1	28.6	0.038	22.5	0.18	15.7	13.3	0.023	3.6
2	0.77	21.9	34.8	0.023	24.1	0.18	9.9	14.7	0.029	4.32
3	0.81	29.1	33	0.012	13.7	0.2	12.7	13.8	0.031	18.6
4	0.35	23.5	34.2	0.009	20.3	0.18	13.7	15.4	0.025	12.1
5	0.82	22.8	26.4	0.021	19.4	0.09	13.4	15	0.03	8.43
6	0.5	14.7	44.7	0.027	15.3	0.06	11.8	12.5	0.017	9.12
7	0.89	21.2	27.6	0.028	16.1	0.22	21	15	0.038	19.3
8	0.9	17.1	37.2	0.023	22.1	0.08	10.4	19.4	0.04	8.76
9	0.77	24.9	40.9	0.030	21.1	0.16	11.3	15	0.023	8.7
10	0.45	21.9	43.5	0.032	17.3	0.2	13.2	16.2	0.029	9.4
Mean	0.71	21.12	35.09	0.024	19.2	0.15	13.31	15.03	0.028	10.2
SD	0.16	3.5	5.1	0.009	3.45	0.05	2.1	1.18	0.005	3.8

## Data Availability

Not applicable.

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
