# Peer review of "Fresh-Water Mollusks as Biomonitors for Ecotoxicity of Nanomaterials"

_nanomaterials, 2021, doi:10.3390/nano11040944_

Round 1
Reviewer 1 Report
In the revised version of the manuscript, the authors made some effort to reduce the similarities with their previous paper (DOI: 10.1039/C8EN00934A). However, the overlaps are still significant. In particular, Table 3 in the submitted paper seems to present the same data as Figure 1 in the previous paper, new Table 1 partially overlaps with the previous Table 1, new Table 2 overlaps with the previous Table 2, and the right panel of Figure 3 is exactly the same as the previous Figure S1. Therefore, the manuscript cannot be recommended for acceptance due to the limited innovativeness.
Author Response
RESPONSES TO REVIEWERS’ COMMENTS
We are thankful to reviewers for your consideration of the manuscript. We revised the paper according to the points raised. All the comments we received on this study have been taken into account in improving the quality of the article, and we present our reply to each of the comments separately.
Comments â„–1:
Manuscript nanomaterials-1089852 by Petr Mashkin et al. reports on the study of ecotoxicity of silicon nanoparticles with three different surface modifications. Even though the topic is interesting, the technical quality of the manuscript is very low. A small number of experiments were conducted, and the ones present in the manuscript are not well-documented by the data. Furthermore, a very similar paper was already published (DOI: 10.1039/C8EN00934A), sharing not only some of the authors but also some of the results. Currently, the other paper is not even cited in the submitted manuscript. Because of this reason, the manuscript cannot be recommended for acceptance. Unless the originality issue is adequately addressed, it is useless to write more detailed comments.
Comments â„–2:
In the revised version of the manuscript, the authors made some effort to reduce the similarities with their previous paper (DOI: 10.1039/C8EN00934A). However, the overlaps are still significant. In particular, Table 3 in the submitted paper seems to present the same data as Figure 1 in the previous paper, new Table 1 partially overlaps with the previous Table 1, new Table 2 overlaps with the previous Table 2, and the right panel of Figure 3 is exactly the same as the previous Figure S1. Therefore, the manuscript cannot be recommended for acceptance due to the limited innovativeness.
Response:
In the revised version of the manuscript, we include our previous and new data of nanotoxicity of various NPs to demonstrate the applicability of described methodology for toxicity testing.
Due to variety of tested NPs, we can demonstrate how the nature of NPs and surface structure significantly influence the toxicity of NPs. It was shown that toxicity of NPs is correlated with the nature of NPs, surface structure, and additives (ligands). In view of size characteristics, we have not found size dependent effects for tested NPs. For some NPs, tested mussels demonstrate moderate sensitivity and animals slightly react to the presence of NPs dispersions (Cu NPs, for instance), in case of earlier studied Si NPs, we observed a significant hazard effect for the same sample. New materials and data were colored in green color.
Reviewer 2 Report
The authors should give more detailed explanations on their modifications and corrections, that made in the revision, corresponding to my concerns in their original submission. The current authors’ responses are not acceptable and very difficult to follow.
Author Response
RESPONSES TO REVIEWERS’ COMMENTS
We are thankful to reviewers for your consideration of the manuscript. We revised the paper according to the points raised. All the comments we received on this study have been taken into account in improving the quality of the article, and we present our reply to each of the comments separately.
Reviewer #1
Comments â„–1:
This manuscript compared the ecotoxicity of three types of Si NPs, C4F9, CH2 and C4H9 using anodonta anatine. From the results of mortality measurement, the authors found that C4F9 are the most toxic with 50% mortality observed within 28 hours. I have the following concerns in this work:
- This work is about ecotoxicity but the authors tried to link it to animal cell toxicity. The authors should mention if there is any relationship between mussel and mouse experiment. It is because all related cell and preclinical model used to employ human cell and small animal, respectively.
- The authors should stress the aim of this work as it is not related to human nanomedicine.
- 2.1: more information about the fabricated NPs should be given such as the particle size distribution, TEM image and x-ray diffraction.
- A schematic diagram to show the differences among C4F9, CH2 and C4H9 should be provided.
- Figure 2: Please label the y and x-axes in the subfigures (channel 1-6)
- The authors did not discuss some important NP variables in the experiment such as particle shape and size. They may have impact on the mussel intake of the NPs.
There is no error analysis and statistical test in this study.
Comments â„–2:
The authors should give more detailed explanations on their modifications and corrections, that made in the revision, corresponding to my concerns in their original submission. The current authors’ responses are not acceptable and very difficult to follow.
This manuscript compared the ecotoxicity of three types of Si NPs, C4F9, CH2 and C4H9 using anodonta anatine. From the results of mortality measurement, the authors found that C4F9 are the most toxic with 50% mortality observed within 28 hours. I have the following concerns in this work:
1. This work is about ecotoxicity, but the authors tried to link it to animal cell toxicity. The authors should mention if there is any relationship between mussel and mouse experiment. It is because all related cell and preclinical model used to employ human cell and small animal, respectively.
Response: New data were added to the article to demonstrate the effect of different NPs toward freshwater mussels and applicability of the described method. Changes were marked in blue color. New materials and data were marked in green color.
- - In the introduction we overviewed research on toxicity of Si NPs towards live species and cell lines as general. There is no correlation between in vitro toxicity to cells and the effect of NPs towards mussels. We described cells toxicity to present the penetration ability of Si NPs and their effect on life cells. Due to limited information on their toxic effects, we did not limit our focus only on ecotoxicity experiments. On the one hand, NPs are promising materials in medical application and, on the other hand, we need to think about their possible toxicity towards human and environment.
2. The authors should stress the aim of this work as it is not related to human nanomedicine.
- Our work is focused on the nondestructive method that can be applied in field studies of ecotoxicity that can be informative at early times of exposure to toxicants to monitor water resources. The described methods were applied to assess toxicity of different NPs (Si, Cu, Ag Cu@Ag) and to study how their properties influence the toxicity.
3. 2.1: more information about the fabricated NPs should be given such as the particle size distribution, TEM image and x-ray diffraction.
- Information about NPs properties and typical TEM picture were included for some NPs (Figures 3-4 and Table 1).
4. A schematic diagram to show the differences among C4F9, CH2 and C4H9 should be provided.
- A schematic diagram was added to show the differences in toxicity and cardiac cycles among silicon NPs with C4F9, CH2 and C4H9 ligands (Figure 7).
5. Figure 2: Please label the y and x-axes in the subfigures (channel 1-6)
- Description was added to Figure 2: The X axis shows the time of the experiment in days. The Y axis shows the amplitude of the heart beats.
6. The authors did not discuss some important NP variables in the experiment such as particle shape and size. They may have impact on the mussel intake of the NPs.
- The influence of key parameters of NPs were discussed and describe It was shown that toxicity of NPs is correlated with the nature of NPs, surface structure, additives. In view of size characteristics, we have not found size-dependent effects for tested NPs, the same for the shape of tested NPs, although they were almost ideal spheres.
There is no error analysis and statistical test in this study.
- Figures were changed and statistical parameters were added on the graphs.
Round 2
Reviewer 1 Report
The authors present a substantially enhanced version of their manuscript studying the effect of various kinds of nanomaterials on fresh-water mollusks. Even though slowly approaching an acceptable state, additional revisions are still necessary, as indicated below.
1. After numerous revisions, the abstract seems to be a group of sentences put in random order rather than a concise text with a clear structure. (This is demonstrated, e.g., by using the term “In conclusion” around half of the abstract length.) Therefore, the abstract should be carefully rewritten.
2. TEM and DLS data should be provided for all the studied nanomaterials (possibly as Supporting Information).
3. In Figure 2, the labels of the x- and y-axis are missing.
4. The abbreviation “Me NPs” is not explained.
5. Terms “in vivo” and “in vitro” should be written in italics.
Author Response
Reviewer #1 (Round 2)
The authors present a substantially enhanced version of their manuscript studying the effect of various kinds of nanomaterials on fresh-water mollusks. Even though slowly approaching an acceptable state, additional revisions are still necessary, as indicated below.
After numerous revisions, the abstract seems to be a group of sentences put in random order rather than a concise text with a clear structure. (This is demonstrated, e.g., by using the term “In conclusion” around half of the abstract length.) Therefore, the abstract should be carefully rewritten.
TEM and DLS data should be provided for all the studied nanomaterials (possibly as Supporting Information).
In Figure 2, the labels of the x- and y-axis are missing.
The abbreviation “Me NPs” is not explained.
Terms “in vivo” and “in vitro” should be written in italics.
Response:
We are very grateful to the reviewer for the efforts in reading carefully our manuscript and giving us very useful comments and recommendations. We revised the paper according to the comments and we present our reply to each of them separately.
The manuscript was revised and necessary changes were done according to recommendations. Changes are made in yellow
- The abstract section has been rewritten and structured.
- TEM and DLS data were included for all tested nanoparticles and added in the Supported information file and in the main text
- Abbreviation for “Me NPs” was added and explained.
- Terms “in vivo” and “in vitro” were presented in italics
Reviewer 2 Report
The authors improved the presentation of the authors’ responses in this third submission, and I can review this work easier. I found that they have made good corrections and modifications in this second revision as per my suggestions. I have no further question.
Author Response
Reviewer #2 (Round 2)
The authors improved the presentation of the authors’ responses in this third submission, and I can review this work easier. I found that they have made good corrections and modifications in this second revision as per my suggestions. I have no further question.
Response: We are grateful to the reviewer for the useful comments that helped us to improve the manuscript.